# Design and Optimization of a Micron-Scale Magnetoelectric Antenna Based on Acoustic Excitation

**DOI:** 10.3390/mi13101584

**Published:** 2022-09-23

**Authors:** Na Li, Xiangyang Li, Bonan Xu, Bin Zheng, Pengchao Zhao

**Affiliations:** Key Laboratory of Electronic Equipment Structure Design, Ministry of Education, Xidian University, Xi’an 710071, China

**Keywords:** antenna miniaturization, ME antenna, mason equivalent circuit model, finite element simulation, high gain

## Abstract

The development of antenna miniaturization technology is limited by the principle of electromagnetic radiation. In this paper, the structure size of the antenna is reduced by nearly two orders of magnitude by using Acoustic excitation instead of electromagnetic radiation. For this magnetoelectric (ME) antenna, the design, simulation and experiment were introduced. Firstly, the basic design theory of magnetoelectric antennas has been refined on a Maxwell’s equations basis, and the structure of the ME antenna is designed by using the Mason equivalent circuit model. The influence mechanism of structure on antenna performance is studied by model simulation. In order to verify the correctness of the proposed design scheme, an antenna sample operating at 2.45 GHz was fabricated and tested. The gain measured is −15.59 dB, which is better than the latest research that has been reported so far. Therefore, the ME antenna is expected to provide an effective new scheme for antenna miniaturization technology.

## 1. Introduction

In recent years, with the rapid development of communication equipment miniaturization, most electronic components have achieved miniaturization. However, the size of an electrically small antenna is still larger than 1/10 of the working wavelength [1,2,3]. What is worse, the impedance matching is difficult, and the radiation efficiency is very low. The core reason is that the traditional small antenna is based on the working principle of the conduction current. Therefore, the inherent ohmic loss of the conduction current results in a reduction in the radiation efficiency. It suffers the platform effect when it is close to the conductive plane, the radiation Q value will increase and the antenna is difficult to match with the impedance [4,5]. Different from the traditional small antenna, the radiation time-varying field of the ME antenna is generated by rotating or oscillating an electric/magnetic dipole moment [6,7,8]. It not only overcomes the ohmic loss but also has a very high-quality factor. It converts the magnetic component of an electromagnetic wave into an acoustic wave and outputs it as a voltage. In turn, the piezoelectric material is voltage-driven to produce strain, which is then transmitted to the magnetic material, triggering it to magnetize and oscillate, and eventually radiates electromagnetic waves [9,10,11]. Based on the working principle of acoustic excitation, the advantages of the ME antenna are: (1) the size of the ME antenna can be reduced to 1/10 or even 1/100 of the size of the traditional small antenna, because the speed of sound waves compared to the speed of electromagnetic waves about five orders of magnitude slower. (2) It radiates without the conduction of an electric current; thus, it solves the problem of low radiation efficiency. (3) Its impedance can be adjusted by changing the sizes of the magnetic and the piezoelectric layers. There is no need to add an external matching network, which solves the problem of impedance matching.

Research of the ME antenna originated from the thin-film composite ME materials [8,12,13,14,15,16]. Greve used MEMS technology to fabricate a cantilever structure by integrating alnfecosib magneto-electromechanical composite thin films onto a silicon substrate and obtained a ME coupling coefficient of up to 737 V/cm∙Oe at a resonant frequency that was 200 times higher than that under non-resonant conditions [17]. However, the further improvement of the magnetoelectric coupling coefficient is limited by the influence of air damping, and the higher the resonant frequency of the antenna, the more significant the damping effect is. Subsequently, the introduction of acoustic resonators solved this problem. On the one hand, the dynamic magnetization of magnetic materials can be excited and controlled by elastic waves based on the magnetoelastic coupling effect. On the other hand, the mechanical energy can be confined in the piezoelectric layer and the magnetostrictive layer through the design of acoustic impedance. Surface and bulk acoustic resonators have very high-quality factors, which creates the basic conditions for improving the magnetoelectric coupling coefficient and enhancing the practicability of magnetoelectric composites [18,19]. In recent years, Yao et al. systematically studied the (bulk acoustic wave) BAW ME antenna by using the finite difference time domain (FDTD) method and, firstly, coupled the Maxwell’s equation and Newton’s equation by using the constitutive relationship of ME materials and constructed a complete mathematical model of the energy, average radiated power and radiated quality factor of the ME antenna [6]. Later, Yao generalized it to the 3D case but only in a simulation [20,21]. Domann et al. further proposed the concept of a strain-powered (SP) antenna and established an electrodynamic analytical model to describe the mechanical coupling of EM radiation of the SP antenna [7]. Nan et al. of Northeastern University (NEU) proposed a nanoelectromechanical system (NEMS) ME antenna excited by acoustic waves [8]. There are two structures: a nanoplate resonator (NPR) and thin-film bulk acoustic resonator (FBAR). However, both of them have the problems of low gain and a narrow bandwidth. Lin further proposed a ME antenna with high magnetic field sensitivity and high gain based on Nan [22]. Schneider et al. further experimentally demonstrated the working principle of near-field multiferroic antennas, and they mainly studied the near field of ME antennas and were not involved the far field [23]. Zaeimbashi proposed a novel ultra-miniaturized wireless implantable device, Nano Neuro RFID (Radio Frequency Identification) [24]. The core of this device is a NPR ME antenna array. They just put forward this design concept and did not really manufacture it, let alone carry out test experiments. Dong et al. applied ME antenna to VLF (Very Low Frequency, 3–30 KHz) [25], and they conducted the near-field testing, not far-field testing. Niu et al. studied a miniaturized low-frequency ME receiving antenna with integrated DC bias, which can achieve a higher performance than existing antennas without a DC bias [26], but they only studied the reception of the antenna, not the transmission of the antenna. Kevin used the Landau–Lifshitz–Gilbert (LLG) equation to accurately analyze the magnetoelastic coupling problem in the ME antenna [27], but they did not consider the spatially dependent electrodynamics governed by Maxwell’s equation. Ren et al. demonstrated the possibility of using only one BAW-actuated ME transducer antenna for communication; however, the simulation method in this work cannot be used for modeling the far field of radiation [28]. Aiming at the problem of low gain and narrow bandwidth of the ME antenna, a new design scheme of the ME antenna is proposed in this paper based on the above works. In this paper, we design and test that, similar to the traditional antenna, the receiving and transmitting processes of the ME antenna are reciprocal. We first reviewed the working principle of the ME antenna and deduce the radiation power formula. For the structural design of the ME antenna, the equivalent circuit method was used to design the thickness of the ME antenna at 2.45 GHz. Based on the coupling of electric field, magnetic field and stress field, we used the finite element method (FEM) to simulate and obtain the far-field radiation pattern of the ME antenna. We tested the gain of the ME antenna to be −15.59 dB, which was better than the gain reported recently.

## 2. The Basic Principle of ME Antenna

The basic principle of the ME antenna is the ME coupling effect, which is the product effect of the piezoelectric effect of piezoelectric materials and the piezomagnetic effect of magnetostrictive materials [29,30].

The working principle is shown in Figure 1. When the antenna transmits EM waves, it applies the RF electric field to the upper and lower sides of the piezoelectric layer of the resonant cavity. The mechanical resonance will generate alternating strain waves, which will then be transmitted to the magnetostrictive layer above.

In the case of the ME antenna, the bulk acoustic resonator utilizes the p-wave mode in the body, and the sound wave propagates along the *Z*-axis. Therefore, under the one-dimensional conditions, the piezoelectric constitutive equation can be rewritten as
(1)$$\begin{array}{l} S_{E}=s_{E}T_{E}+\frac{d_{E}}{\varepsilon_{T}}D\\ E=-\frac{d_{E}}{\varepsilon_{T}}T_{E}+\frac{1}{\varepsilon_{T}}D \end{array}$$

Accordingly, the piezomagnetic constitutive equation can be written as
(2)$$\begin{array}{l} S_{H}=s_{H}T_{H}+\frac{d_{H}}{\mu_{T}}B\\ H=-\frac{d_{H}}{\mu_{T}}T_{H}+\frac{1}{\mu_{T}}B \end{array}$$
where *E* and *H* are the electric and magnetic field intensity vectors, respectively, *D* and *B* are the electric and magnetic flux density vectors, $\varepsilon_{T}\;\mathrm{and}\;\mu_{T}$ are the stress-free permittivity of the piezoelectric layer and stress-free permeability of the magnetostrictive layer, respectively, $s_{E}\;\mathrm{and}\;s_{H}$ are the mechanical compliance constants of piezoelectric layer and magnetostrictive layer, respectively and $d_{E}\;\mathrm{and}\;d_{H}$ are the strain constants of the piezoelectric layer and magnetostrictive layer, respectively. $S_{E}\;\mathrm{and}\;S_{H}$ are the strain field in the piezoelectric layer and magnetostrictive layer, respectively. $T_{E}\;\mathrm{and}\;T_{H}$ are the stress field in the piezoelectric layer and magnetostrictive layer, respectively.

The potential energy in the ME antenna is equal to the sum of the potential energy in the piezoelectric and the piezomagnetic layers. The potential energy in the piezoelectric layer mainly includes mechanical energy in the form of mechanical stress and electrical energy in the form of the electric field:(3)$$\begin{array}{l} W_{PE}=\frac{1}{2}∭S\cdot Tdv+\frac{1}{2}∭D\cdot Edv\\ W_{PM}=\frac{1}{2}∭S\cdot Tdv+\frac{1}{2}∭B\cdot Hdv \end{array}$$

Due to the open-circuit excitation of the piezoelectric layer (D=0) after the initial current pulse drive and the weak magnetic field condition in the magnetostrictive layer (H≈0), it can be derived from the second equations in Equations (1) and (2):(4)$$\begin{array}{l} E=-\frac{d_{E}}{\varepsilon_{T}}T_{E}\\ B=d_{H}T_{H} \end{array}$$

The electromechanical and magnetomechanical coupling figures of merit are given by:(5)$$k_{E}^{2}=\frac{d_{E}^{2}}{s_{E}\varepsilon_{T}},\;k_{H}^{2}=\frac{d_{H}^{2}}{s_{H}\mu_{T}}$$

The mechanical compliance in the magnetic layer and piezoelectric layer are:(6)$$\begin{array}{l} s_{B}=(1-k_{H}^{2})s_{H}\\ s_{D}=(1-k_{E}^{2})s_{E} \end{array}$$

The total energy in the magnetic layer and piezoelectric layer are simplified according to the following:(7)$$\begin{array}{l} W_{PM}=\frac{1}{2}∭S_{H}\cdot T_{H}dv+\frac{1}{2}∭B\cdot Hdv=\frac{1}{2}∭s_{H}T_{H}{}^{2}dv\\ W_{PE}=\frac{1}{2}∭S_{E}\cdot T_{E}dv+\frac{1}{2}∭D\cdot Edv=\frac{1}{2}∭s_{E}\cdot T_{E}{}^{2}dv \end{array}$$

It is assumed that both materials deform at the same time, the strain is equal and the stress is different. The stress in the piezoelectric layer is assumed to be $T_{1}$ and the stress in the piezoelectric layer to be $T_{2}$, so the potential energy stored in the antenna can be expressed as the total potential stored in the form of mechanical stress:(8)$$\begin{array}{l} W_{P}=W_{PE}+W_{PM}=\frac{1}{2}\underset{v_{1}}{∭}s_{E}T_{1}^{2}dv+\frac{1}{2}\underset{v_{2}}{∭}s_{H}T_{2}^{2}dv\\ =\frac{A}{2}s_{E}\int_{z_{1}}T_{1}^{2}dz+\frac{A}{2}s_{H}\int_{z_{2}}T_{2}^{2}dz \end{array}$$

The stress field in the piezoelectric layer is $T_{1}=nT_{0}\sin\left(\frac{2\pi }{\lambda_{ac}}z\right)$, $T_{2}=T_{0}\sin\left(\frac{2\pi }{\lambda_{ac}}z\right)$, $T_{0}$ is the amplitude of the stress field and n is the proportional constant of the stiffness coefficient of the piezoelectric layer and magnetostrictive layer. The device thickness is $d=\frac{\lambda_{ac}}{2}$, and $\lambda_{ac}$ is the wavelength of the sound waves.

By introducing the known stress field function into the potential energy formula, the total potential energy of the stacked structure magnetoelectric antenna can be calculated. In the process of the ME antenna radiating outward, the piezoelectric layer is the driving source of the stress, while the piezomagnetic layer is the radiation region of the antenna, which is mainly responsible for the radiation of the magnetic field coupled by the stress field. Therefore, the expression of the radiated power of a ME antenna is introduced:(9)$$P=\frac{1}{2\eta_{0}}\iint E_{0}^{2}ds$$

In which $E_{0}$ is the aperture electric field formed on the surface of the magnetosphere, and $\eta_{0}$ is the free-space wave impedance.

The radiation power formula of the ME antenna is:(10)$$\begin{array}{l} P=\frac{1}{2\eta_{0}}\iint E_{0}^{2}ds=\frac{\omega^{2}h^{2}}{2\eta_{0}}\iint B^{2}ds=\frac{\omega^{2}h^{2}d_{H}^{2}}{2\eta_{0}}\iint T_{2}^{2}ds\\ =\frac{A\omega^{2}h^{2}d_{H}^{2}T_{0}}{2\eta_{0}}\sin^{2}\left(\frac{2\pi }{\lambda_{ac}}z\right) \end{array}$$

## 3. Design and Impedance Analysis of ME Antenna

The main structure of the ME antenna designed in this paper is based on a cavity-backed FBAR. The substrate is Si, the bottom electrode material is Mo, the piezoelectric material layer is AlN and the magnetostrictive layer is FeGa (also known as the upper electrode). There is an AlN seed layer between the substrate and the lower electrode, and the specific structure and thickness are shown in Figure 2:

In order to determine the resonant frequency and analyze the impedance characteristics of the ME antenna, the Mason equivalent circuit model is used to design the antenna circuit. At first, the structure size and material properties are equivalent to the circuit parameters, and the Mason equivalent circuit models of the piezoelectric layer, magnetostrictive layer and electrode layer are established, respectively, and then, the actual antenna structure is constructed, as shown in Figure 3.

The electrical impedance of the antenna can be simulated by the circuit diagram shown in Figure 4, and the results shown in Figure 4a are the amplitude–frequency characteristic curve of the antenna, and Figure 4b is the phase–frequency characteristic of the antenna impedance curve. It can be seen from the simulation results that the ME antenna has two resonance frequencies—namely, the series resonance frequency is 2.305 GHz, and the parallel resonance frequency is 2.36 GHz.

## 4. Finite Element Simulation and Performance Analysis of ME Antenna

In the previous section, the effect of the antenna structure size and material properties on the electrical characteristics was simulated by using the Mason equivalent circuit model. In order to further analyze the electromagnetic radiation characteristics of the ME antenna, it is necessary to establish an accurate finite element model. As shown in Figure 5, the radius of the resonance region is 100 um, AlN is used as the piezoelectric material and FeGa is used as the magnetostrictive material.

Firstly, the static approximation method is used to study the spatial distribution of stress, strain, displacement and potential, as shown in Figure 6. It can be seen that the maximum stress and strain occur at the interface between the piezoelectric and the magnetostrictive layers. It can be found that the antenna is greatly deformed in the thickness direction, and the maximum displacement can reach 0.25 μm in Figure 6c. From Figure 6d, it can be found that the voltage amplitude at the boundary is significantly higher than that at the middle position.

In order to determine the optimum structural parameters of the antenna, the influence of the structural parameters on the resonant characteristics is analyzed [31].

1.Influence of substrate

For the cavity-backed FBAR structure, the backing is difficult to etch completely during practical machining. Therefore, the influence of the residual substrate thickness on the antenna performance is analyzed. As shown in Figure 7, the resonant frequency of the antenna decreases with the increase of the substrate thickness, because the increase of the substrate thickness directly results in the path of sound wave propagation along the longitudinal direction.

When the antenna substrate thickness H1 is increased from 0.1 μM to 0.7 μM, the real part of the information of admittance is analyzed based on the absolute value of admittance. It can be found that, with the increase of the substrate, more parasitic resonances are generated, the performance of the antenna is deteriorated. Therefore the substrate should be etched as cleanly as possible during actual processing to eliminate the clamping effect of the substrate.

2.Influence of electrode size

As shown in Figure 8, the electrode width has no effect on the operating frequency of the ME antenna, while the larger the electrode width is, the larger the admittance value will be. The increase in area results in an increase in the capacitance of the antenna and reduces the overall impedance of the antenna. Therefore, impedance matching can be performed by adjusting the electrode area. In the meantime, increasing the area of the electrode can reduce the parasitic mode, which can reduce the impact of the energy loss of the acoustic leakage on the antenna efficiency.

3.Influence of electrode shape

In addition to longitudinal resonance, the ME antenna also has transverse parasitic resonance, which can disperse the energy of the main resonance. Therefore, it should be avoided as much as possible. The influence of the electrode shape on parasitic resonance is analyzed in this section. Figure 9 shows a square and a pentagonal electrode. Both of them have the same electrode area, and their admittance curves are shown in Figure 10. It can be seen that the parasitic resonance of the pentagonal electrode is obviously smaller than that of the square electrode. Therefore, the electrode shape of this design is optimized as a pentagonal shape.

Based on the above analysis results, the optimum structural parameters of the magnetoelectric antenna are determined, and the structural model of the magnetoelectric antenna is established. The solid mechanics and low-frequency Maxwell equation are used to calculate the inverse ME effect, and the near-field radiation is obtained. The far-field distribution is shown in Figure 11. In the COMSOL simulation, the input power is set as Pin = 1 W, and it can be calculated that the real gain Greal = 1.93 × 10^−13^. Since impedance matching is not considered in the 3D model, the S11 of the resonance point is −0.0183 dB. Excluding the mismatch factor, the radiation efficiency can be calculated to be about 2.55 × 10^−11^.

## 5. Fabrication and Testing of Antenna Samples

In order to verify the superiority of the proposed design scheme, a 2.45-GHz antenna sample is fabricated, and its radiation performance is tested. The processing flow is shown in Figure 12.

The optical image of the ME antenna is shown in Figure 13.

The test platform is shown in Figure 14.

The reflection coefficient (S11) of the antenna is measured as shown in Figure 15. It can be obtained that the resonant frequency of the antenna is about 2.49 GHz, and the peak return loss is −17.3 dB.

Then, the S11 and S21 curves are tested in the anechoic chamber environment, where S21 indicates that the ME antenna transmits the signal, and the horn antenna receives the signal. The S11 curve is converted into the Z11 curve and put together with the S21 curve, as shown in Figure 16. The test results show that the antenna has an obvious radiation enhancement at the parallel resonance point. It indicates that the ME antenna produces obvious radiation at the mechanical resonance frequency, which verifies that the radiation of the ME antenna comes from the ME coupling of the mechanical resonance.

The S12 and S21 parameter curves in Figure 17 are basically the same, indicating that the ME antenna has reciprocity under the action of the external magnetic field of small signals. The ME antenna meets the reciprocity principle of traditional antenna.

1.Calculation of Antenna Gain

Measured against the antenna in an anechoic chamber at the resonant frequency of 2.49 GHz, the $S_{21,a}$ is −50.42 dB, and the $S_{21,r}$ is −18.03 dB, wherein the gain $G_{r}$ of the reference standard horn antenna is 16.8 dB, and $S_{21,a}$ and $S_{21,r}$ represent the S21 peak value of the ME antenna to be tested and the reference horn antenna test, respectively. According to the comparison method, the antenna gain is calculated to be −15.59 dB.

2.Antenna pattern

In order to fully analyze the radiation characteristics of the ME antenna, its pattern is tested. Verified by the literature, the radiation of the ME antenna can be equivalent to a model of a magnetic dipole. Using the MATLAB simulation, the radiation pattern of the equivalent magnetic dipole of the ME antenna can be obtained, as shown in Figure 18. Further radiation power P is calculated as:$2.61\times 10^{-8}W$.

## 6. Conclusions

The proposed ME antenna provided a new method for antenna miniaturization. In this paper, a ME antenna structure was designed, the finite element simulation was carried out on it and the sample was prepared and tested. The results showed that the radiation of the ME antenna originates from the mechanical resonance. It also shows that the ME antenna has the potential to solve the problems of difficult miniaturization and impedance matching of traditional antennas. It can be equivalent to a dipole antenna, its radiation signal comes from the ME coupling and its gain is measured to be −15.59 dB.

## Figures and Tables

**Figure 1 micromachines-13-01584-f001:**
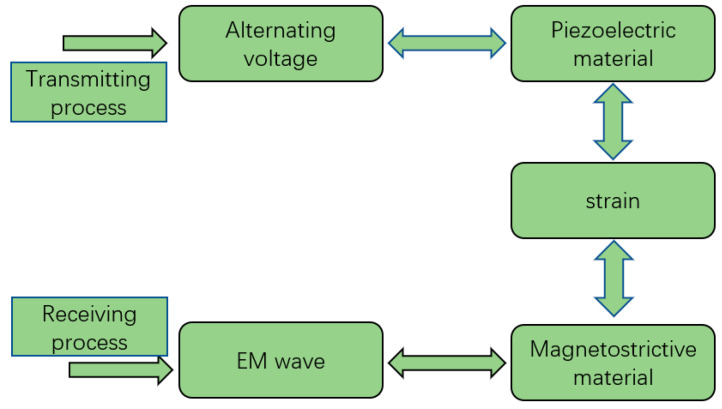
The working principle of the ME antenna.

**Figure 2 micromachines-13-01584-f002:**
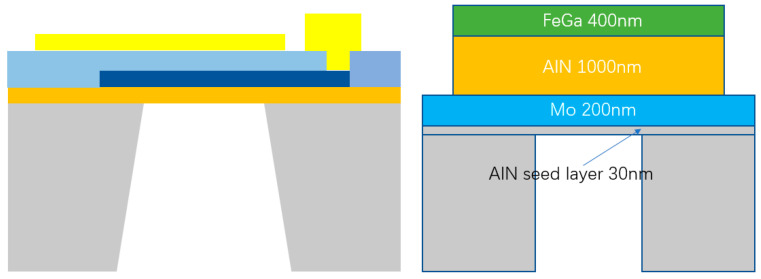
The left is the cavity-backed FBAR; the right is the structure and materials of the ME antenna.

**Figure 3 micromachines-13-01584-f003:**
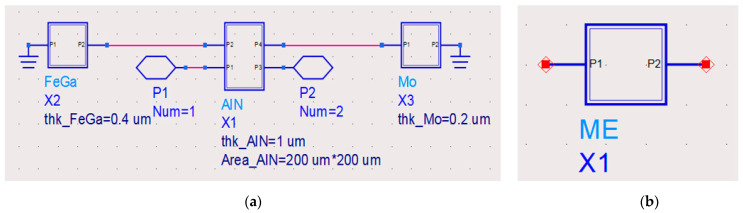
(**a**) The cascade diagram of the ME antenna, and (**b**) the ME antenna encapsulation.

**Figure 4 micromachines-13-01584-f004:**
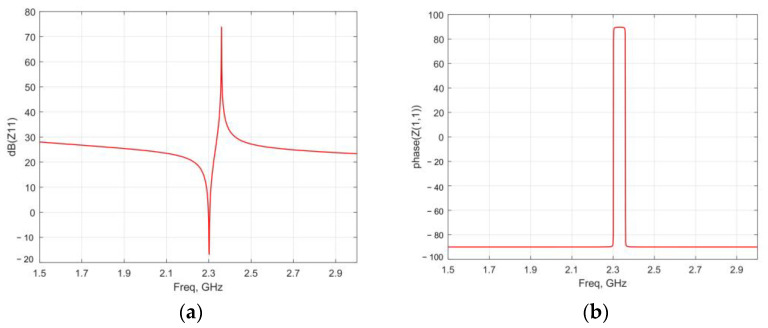
Amplitude (**a**) and phase (**b**) of the antenna impedance.

**Figure 5 micromachines-13-01584-f005:**
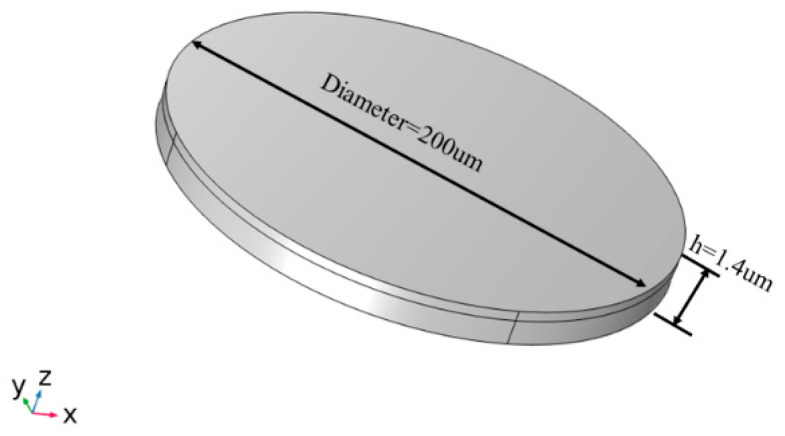
Antenna model.

**Figure 6 micromachines-13-01584-f006:**
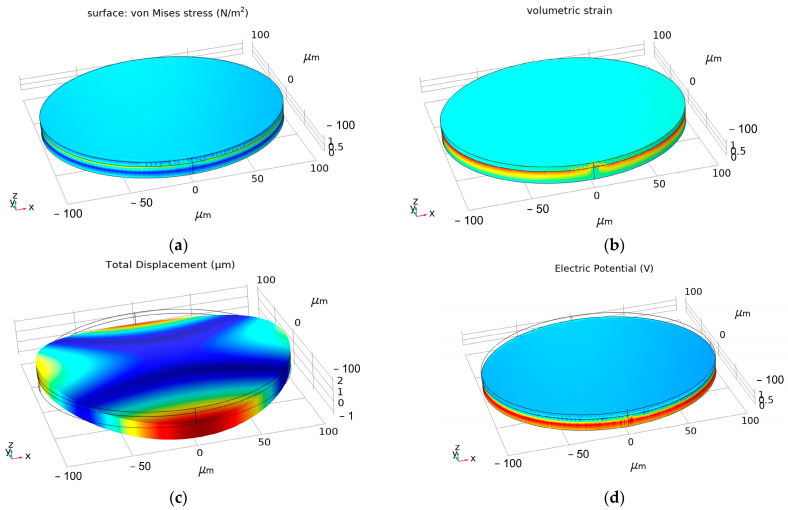
Distribution of the structure and electrical parameters of the ME antenna under a magnetic field. (**a**) Stress, (**b**) strain, (**c**) displacement and (**d**) electric potential.

**Figure 7 micromachines-13-01584-f007:**
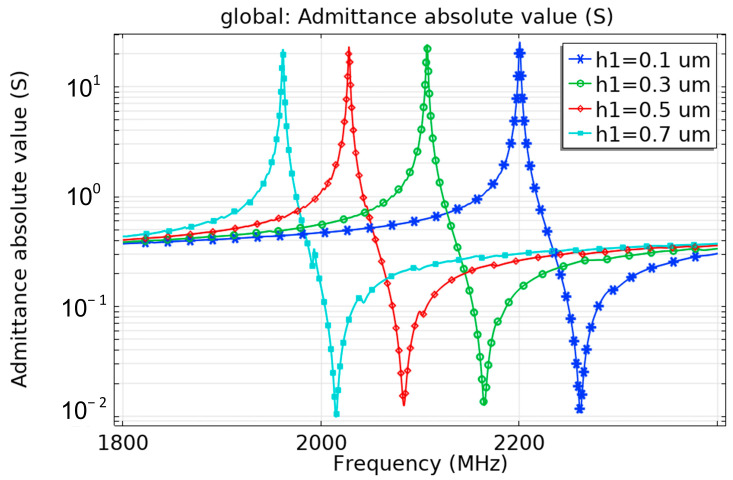
Influence of substrate thickness h on the admittance curve.

**Figure 8 micromachines-13-01584-f008:**
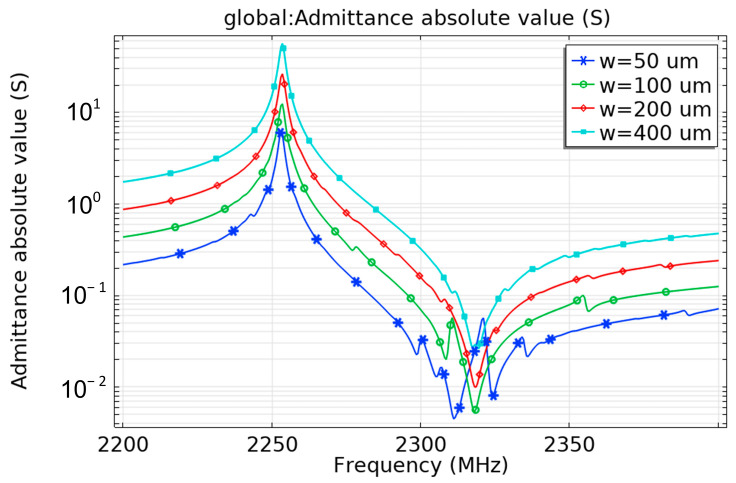
Influence of electrode width on the admittance curve.

**Figure 9 micromachines-13-01584-f009:**
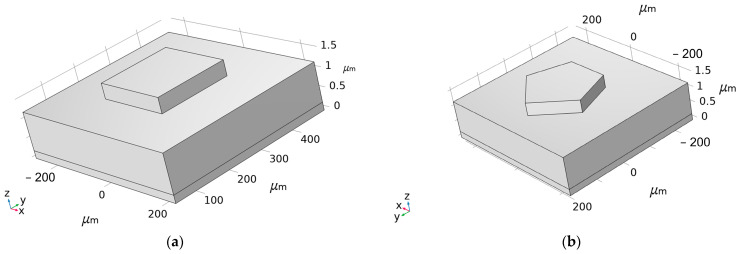
Three-dimensional models of different electrode shapes: (**a**) square electrode and (**b**) pentagonal electrode.

**Figure 10 micromachines-13-01584-f010:**
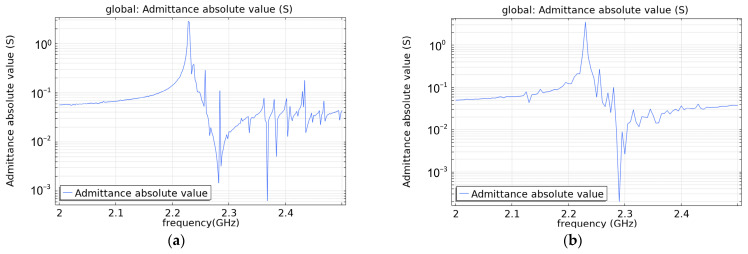
Absolute values of admittance for different electrode shapes: (**a**) square electrode and (**b**) pentagonal electrode.

**Figure 11 micromachines-13-01584-f011:**
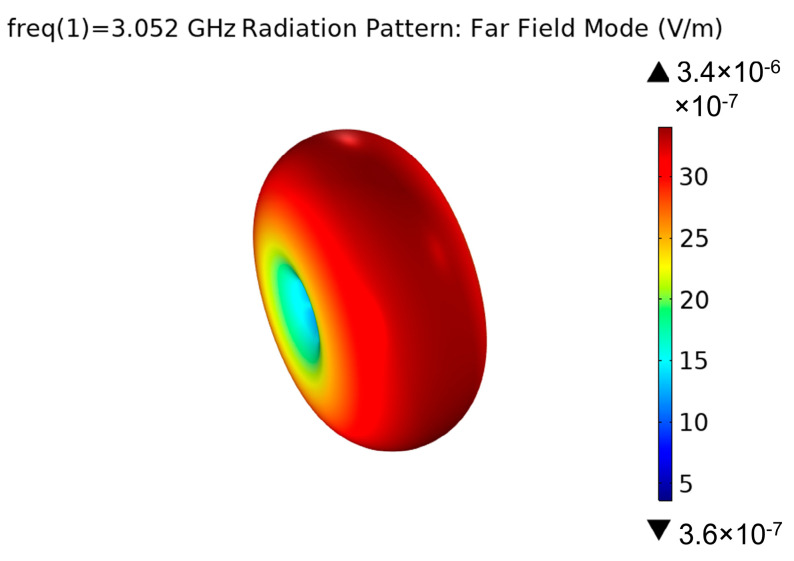
Far-field 3D pattern of the antenna.

**Figure 12 micromachines-13-01584-f012:**
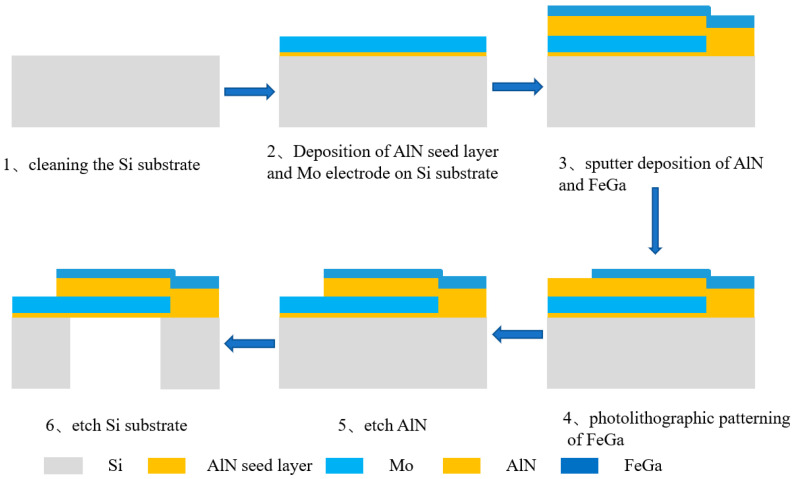
ME antenna preparation process.

**Figure 13 micromachines-13-01584-f013:**
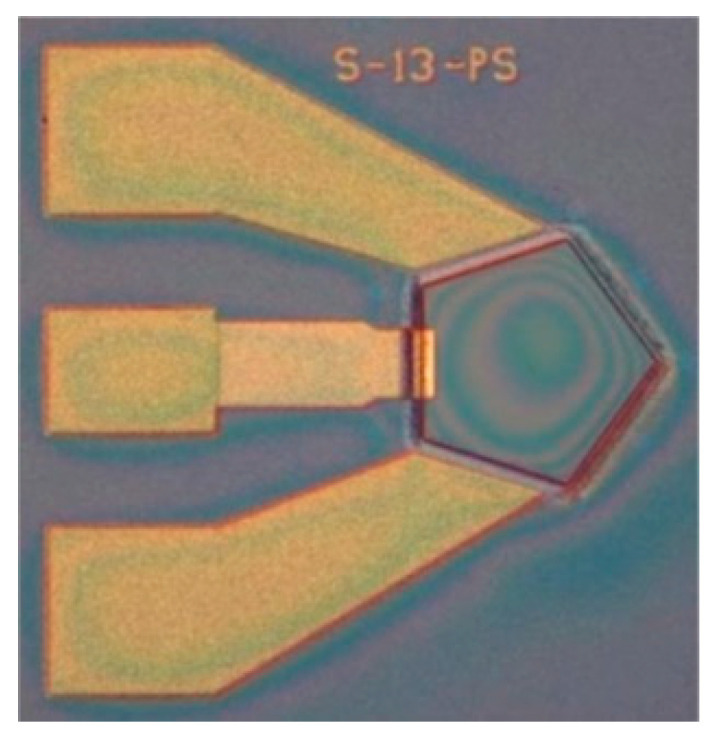
Optical image of the ME antenna.

**Figure 14 micromachines-13-01584-f014:**
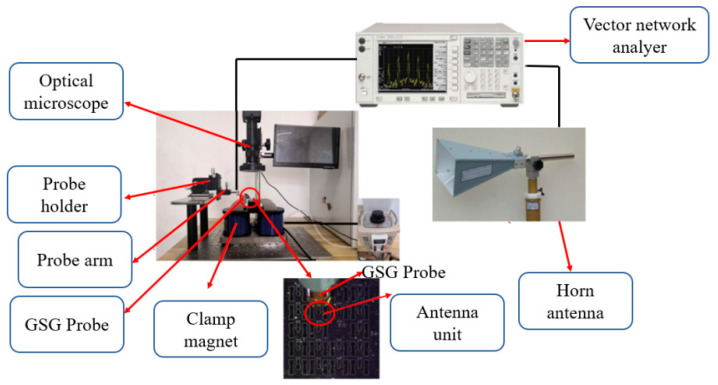
Testing platform.

**Figure 15 micromachines-13-01584-f015:**
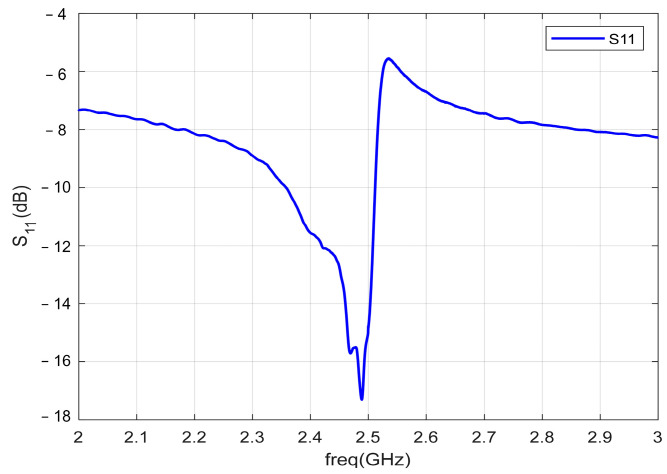
S11 curve of the ME antenna.

**Figure 16 micromachines-13-01584-f016:**
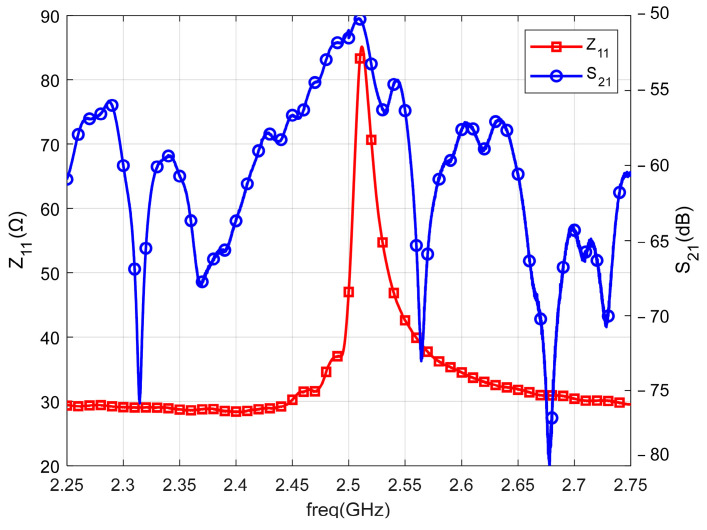
Z11 and S21 parameters of the antenna.

**Figure 17 micromachines-13-01584-f017:**
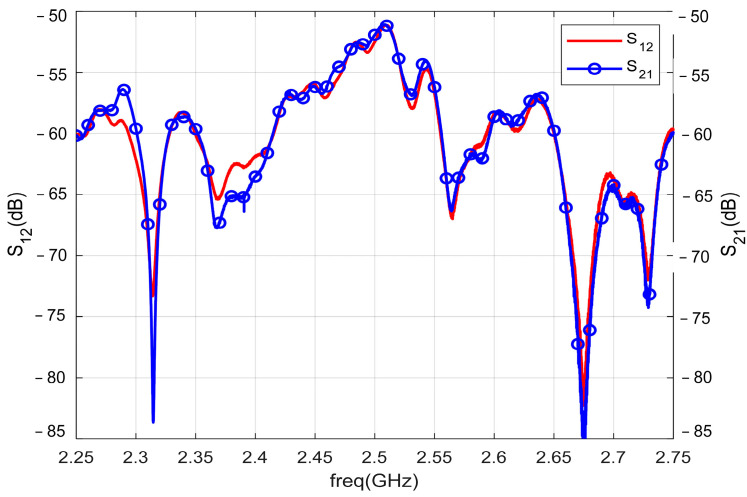
S12 and S21 parameters of the ME antenna.

**Figure 18 micromachines-13-01584-f018:**
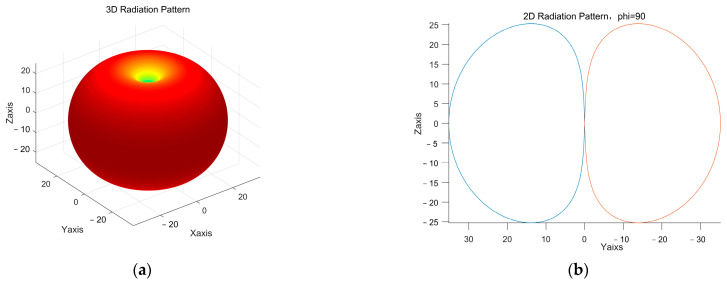
Pattern of the ME antenna: (**a**) 3D radiation pattern and (**b**) principal gain (phi = 90) pattern.

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
