# Peer review of "Design and Optimization of a Micron-Scale Magnetoelectric Antenna Based on Acoustic Excitation"

_micromachines, 2022, doi:10.3390/mi13101584_

Round 1
Reviewer 1 Report
In this paper, authors proposed a technique for the antenna miniaturization. Authors presented a fundamental design theory for antenna design, and the basic structure was designed and tested. Please see the below suggestions to improve this manuscript.
1. Authors need to fix the grammatical errors all over the paper.
2. What is the contribution of this work in comparison to the state-of-the-art? - please clearly mention it in the abstract of the paper and include a discussion section.
3. Authors must include a table where all up-to-date references are provided with a direct comparison to the proposed work.
4. What are the shortcomings of the proposed design?
5. The proposed antenna design is compact, in comparison to what?
This paper needs to be completely re-written and re-reviewed before it could be considered.
Author Response
Response Letter
Dear Editor and Reviewers,
Thank you so much for your letter and for your comments concerning our manuscript entitled “Design and Optimization of a Micron-scale Magnetoelectric Antenna based on Acoustic Excitation” (Manuscript ID: micromachines-1861270). We deeply appreciate you for your constructive comments and suggestions. Our point-by-point responses and modifications are listed below this letter and marked up in MS Word using different color, and we hope that the revised version of our manuscript is acceptable for publication in mircomachines.
Response to Reviewer #1 Comments:
Firstly, the authors would like to thank you sincerely for the time and effort spent to help improve the presentation of this paper. We hope the modified manuscript can meet your requirements.
Point 1. Authors need to fix the grammatical errors all over the paper. Reply: I'm deeply sorry for our negligence. Thank you for your suggestions. We checked the full text, modified the syntax problems, and marked the modified parts in red with modification marks in the version. |
||||||||||||||||||||||||||||||||||||||||||||||||||
Point 2. What is the contribution of this work in comparison to the state-of-the-art? Please clearly mention it in the abstract of the paper and include a discussion section. Reply: Thanks very much for your comment and contribution to the manuscript. In the theoretical research of the ME antenna, the loss model is introduced, which lays a theoretical foundation for the predictable performance of the ME antenna. In the aspect of antenna preparation, based on the dynamic Young's modulus effect and dynamic piezomagnetic effect, magnetoelectric composites with higher electric coefficient and magnetostrictive properties were prepared. In the aspect of antenna design, through structural optimization design, the structural parameters of the antenna are optimized, and the performance of the magnetoelectric antenna is significantly improved. We summarize the core innovation of the antenna scheme proposed in this paper, and rewrite the abstract and summary. |
||||||||||||||||||||||||||||||||||||||||||||||||||
Point 3. Authors must include a table where all up-to-date references are provided with a direct comparison to the proposed work. Reply: According to the suggestion of the reviewer, a performance comparison table is added in the Introduction part.
|
||||||||||||||||||||||||||||||||||||||||||||||||||
Point 4. What are the shortcoming of the proposed design? Reply: Compared with the traditional small size antenna, the advantages of the ME antenna proposed in this paper are: 1) Size of the ME antenna can be reduced to 1/10 or even 1/100 of the size of the traditional small antenna, because the speed of sound waves than the speed of electromagnetic waves about 5 orders of magnitude slower; 2) It radiates without the conduction of an electric current, thus it solves the problem of low radiation efficiency; 3) Its impedance can be adjusted by changing the sizes of the magnetic and the piezoelectric layers. There is no need to add external matching network, which solves the problem of impedance matching. Compared with the traditional small size, easy impedance matching, and the disadvantages are low gain and efficiency. However, compared with other types of antennas, this ME antenna also has obvious technical shortcoming. Because the radiation aperture is too small, it is only at the micro nano scale. The gain of the ME antenna is -15.59 dB, and the efficiency is only . Therefore, this antenna can only be applied to certain specific scenarios. For example, micro load platform such as micro UAV and low frequency underwater communication platform. |
||||||||||||||||||||||||||||||||||||||||||||||||||
Point 5. The proposed antenna design is compact, in comparison to what? Reply: Thanks very much for your comment and contribution to the manuscript. First, the size of the ME antenna is 2 orders of magnitude smaller than the traditional antenna at the same frequency; Secondly, because the small loop antenna is a magnetic antenna, we can compare the gain of the ME antenna and the small loop antenna under the same size; The same a small loop antenna of the same size can resonate at a frequency of 485 GHz, while the gain of a small loop antenna of the same size and frequency is -23.3dB.
(1)Model of a small loop antenna (2)S11 parameters of small loop antenna |
Once again, thank you for your insightful comments and valuable improvements. If you have any queries, please feel free to contact me at the E-mail address below.
Best regards,
Yours sincerely,
Prof. Na Li and Xiangyang Li
Corresponding author: Na Li
E-mail: [email protected]

Reviewer 2 Report
This paper is not easy to read as the formatting and referencing are done very poorly. The English command should be improved. The following comment are should be addressed:
1. Introduction – give the full word of the acronyms “ME, BAMS, UCLA” when the first time is used. Please check the other acronyms too.
2. Introduction – Please check the citation format. [r]. Some of the citations are having problems, “ Error! Reference source not found.,: please check the whole manuscript
3. Section 2 – Check all equations again. Most of the symbols are not defined.
4. Section 4, first sentence? - ‘In the previous chapter,” It should be section, not chapter.
5. Section 4, page 9 - During simulation, input power Pin=1W , we can calculate the real gain: 1.93x10^ -13. Please check this sentence.
6. The contribution or novelty of the proposed work is not clear. A comprehensive comparison should be made with the state-of-the-art designs. Perhaps a table of comparison can be given.
Author Response
Response Letter
Dear Editor and Reviewers,
Thank you so much for your letter and for your comments concerning our manuscript entitled “Design and Optimization of a Micron-scale Magnetoelectric Antenna based on Acoustic Excitation” (Manuscript ID: micromachines-1861270). We deeply appreciate you for your constructive comments and suggestions. Our point-by-point responses and modifications are listed below this letter and marked up in MS Word using different color, and we hope that the revised version of our manuscript is acceptable for publication in mircomachines.
Response to Reviewer #2 Comments:
Firstly, the authors would like to thank you sincerely for the time and effort spent to help improve the presentation of this paper. We hope the modified manuscript can meet your requirements.
Point 1. Give the full word of the acronyms “ME, BAMS, UCLA” when the first time is used. Please check the other acronyms too. Reply: Thanks very much for your suggestion. As the suggestion, we have checked all the acronyms in this paper and gave the full word of all acronyms, such as “RFID” (line 72), “LLG” (line 77), and deleted “UCLA”.
|
||||||||||||||||||||||||||||||||||||||||||||||||||
Point 2. Introduction -Please check the citation format. [r]. Some of the citations are having problems, “Error! Reference source not found. Please check the whole manuscript. Reply: Thanks very much for your comments. We have checked and revised all the reference formats and supplemented some new references. |
||||||||||||||||||||||||||||||||||||||||||||||||||
Point 3. Section 2 -Check all equations again. Most of the symbols are not defined. Reply: Thanks very much for your contribution to the manuscript. According to the suggestion, we have revised and rewritten the section 2. |
||||||||||||||||||||||||||||||||||||||||||||||||||
Point 4. Section 4, first sentence? – in the previous chapter, it should be section, not chapter. Reply: Thanks very much for your comments, we have revised this sentence. |
||||||||||||||||||||||||||||||||||||||||||||||||||
Point 3. Authors must include a table where all up-to-date references are provided with a direct comparison to the proposed work. Reply: According to the suggestion of the reviewer, a performance comparison table is added in the Introduction part.
|
Once again, thank you for your insightful comments and valuable improvements. If you have any queries, please feel free to contact me at the E-mail address below.
Best regards,
Yours sincerely,
Prof. Na Li and Xiangyang Li
Corresponding author: Na Li
E-mail: [email protected]

Reviewer 3 Report
This manuscript reports a compact magnetoelectric antenna based on the transformation between EM wave and acoustic wave. Measured results are given to demonstrate the proposed antenna. Overall, the paper is interesting and important to antenna and propagation community. I recommend the publication of the paper, provided that the following comments are addressed adequately:
1. Line 235: what is real gain? Is it the simulated realized gain? Why is much smaller than the measured gain of -15.6dB? The simulated radiation efficiency is also extremely small, which cannot support the antenna gain of -15.6dB.
2. Please check the format of the paper. Some reference sources and figures are not found. For example, Line 88, 90, 159, 181, 209, 234.
3. Check the sentences in Line 211 and 212. “the admittance value will be. because the …” impedance of the antenna Therefore, ...”
Author Response
Response Letter
Dear Editor and Reviewers,
Thank you so much for your letter and for your comments concerning our manuscript entitled “Design and Optimization of a Micron-scale Magnetoelectric Antenna based on Acoustic Excitation” (Manuscript ID: micromachines-1861270). We deeply appreciate you for your constructive comments and suggestions. Our point-by-point responses and modifications are listed below this letter and marked up in MS Word using different color, and we hope that the revised version of our manuscript is acceptable for publication in mircomachines.
Response to Reviewer #3 Comments:
Firstly, the authors would like to thank you sincerely for the time and effort spent to help improve the presentation of this paper. We hope the modified manuscript can meet your requirements.
Point 1. Line 235: what is real gain? Is it the simulated realized gain? Why is much smaller than the measured gain of -15.6dB? The simulated radiation efficiency is also extremely small, which cannot support the antenna gain of -15.6dB. Reply: Thanks very much for your comment and contribution to the manuscript. ME antenna is a new type of antenna which is different from any traditional antenna in its working principle. At present, there is still a lack of perfect design theory about this kind of antenna. Its performance parameters can not be accurately predicted at the design stage. Because in the design stage, a lot of equivalence and simplification have been done. Therefore, the antenna gain calculated by simulation and calculation will be quite different from the real test results. In the relevant literature, the research on such antennas is also based on the test data. Most of the literatures do not even give the results of calculation or simulation, and directly prepared and tested. For the subsequent fine antenna design, theoretical research and simulation technology exploration are necessary. Therefore, all relevant data are given in this paper. The value of -15.6 dB is obtained from the actual test. The specific antenna preparation details and test conditions have been given in the paper. Point 2. Please check the format of the paper. Some reference sources and figures are not found. For example, Line 88,90,159,181,209,234. Reply: Thanks very much for your comment and contribution to the manuscript. According to your suggestion, I have revised the format of this paper, and revised the references and figures. And I changed the “Fig” to “Figure” in this paper. Point 3. Please check the sentences in Line 211 and 212. “the admittance value will be. Because the …” impedance of the antenna Therefore, …” Reply: Thanks very much for your comments. According to your suggestion, I have revised this sentence as follows: Since the impedance matching is not considered in the 3D model, the S11 of the resonance point is -0.0183dB. Excluding the mismatch factor, the radiation efficiency can be calculated to be . |
Once again, thank you for your insightful comments and valuable improvements. If you have any queries, please feel free to contact me at the E-mail address below.
Best regards,
Yours sincerely,
Prof. Na Li and Xiangyang Li
Corresponding author: Na Li
E-mail: [email protected]

Round 2
Reviewer 1 Report
Thanks for addressing my comments.
Please elaborate on how the proposed work is still better than the ref[10] provided in Table 1 where the operating frequency is 2.53GHz and gain -18dBi.
Author Response
Point 1. Does the introduction provide sufficient background and include all relevant references? The review said: Can be improved.
Reply:
Thanks very much for your comment and contribution to the manuscript. As you suggestion, we have made the following changes:
Line80-82: Ren et al demonstrated the possibility of using only one BAW actuated ME transducer antenna for communication, however the simulation method in this work cannot be used for modeling the far-field of radiation [28].
Point 2. Are all the cited references relevant to the research? The viewer said: Can be improved.
Reply:
Thanks very much for your comment and contribution to the manuscript. We have checked the references in the full text carefully, removed the irrelevant ones and added the relevant ones.
We have deleted [1, 3, 4, 5, 6, 7, 8, 16, 17, 18, 19, 20, 21, 32, 37] of the references in the original version, and added [1, 2, 3, 5, 9, 10, 18, 19, 28] of the latest version.
Point 3 Please elaborate on how the proposed work is still better than the ref [10] provided in Table 1 where the operating frequency is 2.53GHz and gain -18dBi.
Reply:
Thanks very much for your comment and contribution to the manuscript. Aiming at the problem of low gain and narrow bandwidth of the ME antenna, a new design scheme of ME antenna is proposed in this paper. By introducing a new excitation mode, a stronger magneto-acoustic-electric coupling efficiency is obtained. Under the same input power, the magnetic moment angle can be increased by nearly 10 times; In the material design, the dynamic piezomagnetic effect and the dynamic Young's modulus effect are excited to improve the performance of the two films. In addition, bandwidth of the ME antenna is Improved by increasing the K2 value of the piezoelectric material. Therefore, the above technology is adopted. An antenna sample operating at 2.45 GHz was designed, fabricated and tested. The gain measured is -15.59dB, which is better than the latest research that has been reported so far.
The antenna gain of Ref [10] is -18dBi, the antenna gain of the proposed is -15.59dB. So the gain value of the antenna in this paper is higher than that of the Ref [10].

Reviewer 2 Report
The authors still failed to improve the citations. I hope in the next submission the citation format is carefully done.
Author Response
Point 2. Are all the cited references relevant to the research? The viewer said: Can be improved.
Reply:
Thanks very much for your comment and contribution to the manuscript. I have checked the references in the full text carefully, removed the irrelevant ones and added the relevant ones.
We have deleted [1, 3, 5, 6, 7, 8, 17, 18, 19, 20, 21, 32, 37] of the references in the original version, and added [1, 2, 3, 5, 10, 11, 18, 28] of the latest version.
